# An Embedding Framework for Consistent Polyhedral Surrogates

**Jessie Finocchiaro**
jefi8453@colorado.edu
CU Boulder

**Rafael Frongillo**
raf@colorado.edu
CU Boulder

**Bo Waggoner**
bwag@colorado.edu
CU Boulder

## Abstract

We formalize and study the natural approach of designing convex surrogate loss functions via embeddings for problems such as classification or ranking. In this approach, one embeds each of the finitely many predictions (e.g. classes) as a point in $\mathbb{R}^d$, assigns the original loss values to these points, and convexifies the loss in some way to obtain a surrogate. We prove that this approach is equivalent, in a strong sense, to working with polyhedral (piecewise linear convex) losses. Moreover, given any polyhedral loss $L$, we give a construction of a link function through which $L$ is a consistent surrogate for the loss it embeds. We go on to illustrate the power of this embedding framework with succinct proofs of consistency or inconsistency of various polyhedral surrogates in the literature.

## 1 Introduction

Convex surrogate losses are a central building block in machine learning for classification and classification-like problems. A growing body of work seeks to design and analyze convex surrogates for given loss functions, and more broadly, understand when such surrogates can and cannot be found. For example, recent work has developed tools to bound the required number of dimensions of the surrogate's hypothesis space [13, 24]. Yet in some cases these bounds are far from tight, such as for *abstain loss* (classification with an abstain option) [4, 24, 25, 33, 34]. Furthermore, the kinds of strategies available for constructing surrogates, and their relative power, are not well-understood.

We augment this literature by studying a particularly natural approach for finding convex surrogates, wherein one "embeds" a discrete loss. Specifically, we say a convex surrogate $L$ embeds a discrete loss $\ell$ if there is an injective embedding from the discrete reports (predictions) to a vector space such that (i) the original loss values are recovered, and (ii) a report is $\ell$-optimal if and only if the embedded report is $L$-optimal. If this embedding can be extended to a calibrated link function, which maps approximately $L$-optimal reports to $\ell$-optimal reports, then consistency follows [2]. Common examples of this general construction include hinge loss as a surrogate for 0-1 loss and the abstain surrogate mentioned above.

Using tools from property elicitation, we show a tight relationship between such embeddings and the class of polyhedral (piecewise-linear convex) loss functions. In particular, by focusing on Bayes risks, we show that every discrete loss is embedded by some polyhedral loss, and every polyhedral loss function embeds some discrete loss. Moreover, we show that any polyhedral loss gives rise to a calibrated link function to the loss it embeds, thus giving a very general framework to construct consistent convex surrogates for arbitrary losses.

**Related works.** The literature on convex surrogates focuses mainly on smooth surrogate losses [4, 5, 7, 8, 26, 30]. Nevertheless, nonsmooth losses, such as the polyhedral losses we consider, have been proposed and studied for a variety of classification-like problems [19, 31, 32]. A notable addition to this literature is Ramaswamy et al. [25], who argue that nonsmooth losses may enable

dimension reduction of the prediction space (range of the surrogate hypothesis) relative to smooth losses, illustrating this conjecture with a surrogate for *abstain loss* needing only $\log n$ dimensions for $n$ labels, whereas the best known smooth loss needs $n - 1$. Their surrogate is a natural example of an embedding (cf. Section 5.1), and serves as inspiration for our work.

While property elicitation has by now an extensive literature [10, 12, 15, 17, 18, 22, 28, 29], these works are mostly concerned with point estimation problems. Literature directly connecting property elicitation to consistency is sparse, with the main reference being Agarwal and Agarwal [2]; note however that they consider single-valued properties, whereas properties elicited by general convex losses are necessarily set-valued.

## 2 Setting

For discrete prediction problems like classification, due to hardness of directly optimizing a given discrete loss, many machine learning algorithms can be thought of as minimizing a surrogate loss function with better optimization qualities, e.g., convexity. Of course, to show that this surrogate loss successfully addresses the original problem, one needs to establish consistency, which depends crucially on the choice of link function that maps surrogate reports (predictions) to original reports. After introducing notation, and terminology from property elicitation, we thus give a sufficient condition for consistency (Def. 4) which depends solely on the conditional distribution over $\mathcal{Y}$.

### 2.1 Notation and Losses

Let $\mathcal{Y}$ be a finite outcome (label) space, and throughout let $n = |\mathcal{Y}|$. The set of probability distributions on $\mathcal{Y}$ is denoted $\Delta_{\mathcal{Y}} \subseteq \mathbb{R}^{\mathcal{Y}}$, represented as vectors of probabilities. We write $p_y$ for the probability of outcome $y \in \mathcal{Y}$ drawn from $p \in \Delta_{\mathcal{Y}}$. We first discuss the conditional setting, with just labels $\mathcal{Y}$ and no features $\mathcal{X}$, and show in § 2.3 how these notions relate to the usual $\mathcal{X} \times \mathcal{Y}$ setting.

We assume that a given discrete prediction problem, such as classification, is given in the form of a *discrete loss* $\ell : \mathcal{R} \to \mathbb{R}_+^{\mathcal{Y}}$, which maps a report (prediction) $r$ from a finite set $\mathcal{R}$ to the vector of loss values $\ell(r) = (\ell(r)_y)_{y \in \mathcal{Y}}$ for each possible outcome $y \in \mathcal{Y}$. We will assume throughout that the given discrete loss is *non-redundant*, meaning every report is uniquely optimal (minimizes expected loss) for some distribution $p \in \Delta_{\mathcal{Y}}$. Similarly, surrogate losses will be written $L : \mathbb{R}^d \to \mathbb{R}_+^{\mathcal{Y}}$, typically with reports written $u \in \mathbb{R}^d$. We write the corresponding expected loss when $Y \sim p$ as $\langle p, \ell(r) \rangle$ and $\langle p, L(u) \rangle$. The *Bayes risk* of a loss $L : \mathbb{R}^d \to \mathbb{R}_+^{\mathcal{Y}}$ is the function $\underline{L} : \Delta_{\mathcal{Y}} \to \mathbb{R}_+$ given by $\underline{L}(p) := \inf_{u \in \mathbb{R}^d} \langle p, L(u) \rangle$; naturally for discrete losses we write $\underline{\ell}$ (and the infimum is over $\mathcal{R}$).

For example, 0-1 loss is a discrete loss with $\mathcal{R} = \mathcal{Y} = \{-1, 1\}$ given by $\ell_{0\text{-}1}(r)_y = \mathbb{1}\{r \neq y\}$, with Bayes risk $\underline{\ell}_{0\text{-}1}(p) = 1 - \max_{y \in \mathcal{Y}} p_y$. Two important surrogates for $\ell_{0\text{-}1}$ are hinge loss $L_{\text{hinge}}(u)_y = (1 - yu)_+$, where $(x)_+ = \max(x, 0)$, and logistic loss $L(u)_y = \log(1 + \exp(-yu))$ for $u \in \mathbb{R}$.

Most of the surrogates $L$ we consider will be *polyhedral*, meaning piecewise linear and convex; we therefore briefly recall the relevant definitions. In $\mathbb{R}^d$, a *polyhedral set* or *polyhedron* is the intersection of a finite number of closed halfspaces. A *polytope* is a bounded polyhedral set. A convex function $f : \mathbb{R}^d \to \mathbb{R}$ is *polyhedral* if its epigraph is polyhedral, or equivalently, if it can be written as a pointwise maximum of a finite set of affine functions [27].

**Definition 1** (Polyhedral loss). *A loss $L : \mathbb{R}^d \to \mathbb{R}_+^{\mathcal{Y}}$ is* polyhedral *if $L(u)_y$ is a polyhedral (convex) function of $u$ for each $y \in \mathcal{Y}$.*

For example, hinge loss is polyhedral, whereas logistic loss is not. To motivate our focus on polyhedral losses, we echo Ramaswamy et al. [25, Section 1.2], who note that smooth surrogates often encode much more information than necessary, and in these cases non-smooth surrogates are the only candidates to achieve a low dimension $d$ above.

### 2.2 Property Elicitation

To make headway, we will appeal to concepts and results from the property elicitation literature, which elevates the *property*, or map from distributions to optimal reports, as a central object to study in its own right. In our case, this map will often be multivalued, meaning a single distribution could

yield multiple optimal reports. (For example, when $p = (1/2, 1/2)$, both $r = 1$ and $r = -1$ optimize 0-1 loss.) To this end, we will use double arrow notation to mean a mapping to all nonempty subsets, so that $\gamma : \Delta_{\mathcal{Y}} \rightrightarrows \mathcal{R}$ is shorthand for $\Gamma : \Delta_{\mathcal{Y}} \to 2^{\mathcal{R}} \setminus \emptyset$. See the discussion following Definition 3 for conventions regarding $\mathcal{R}, \Gamma, \gamma, L, \ell$, etc.

**Definition 2** (Property, level set). *A property is a function $\Gamma : \Delta_{\mathcal{Y}} \rightrightarrows \mathcal{R}$. The* level set *of $\Gamma$ for report $r$ is the set $\Gamma_r := \{p : r \in \Gamma(p)\}$.*

Intuitively, $\Gamma(p)$ is the set of reports which should be optimal for a given distribution $p$, and $\Gamma_r$ is the set of distributions for which the report $r$ should be optimal. For example, the *mode* is the property $\mathrm{mode}(p) = \arg\max_{y \in \mathcal{Y}} p_y$, and captures the set of optimal reports for 0-1 loss: for each distribution over the labels, one should report the most likely label. In this case we say 0-1 loss *elicits* the mode, as we formalize below.

**Definition 3** (Elicits). *A loss $L : \mathcal{R} \to \mathbb{R}_+^{\mathcal{Y}}$, elicits a property $\Gamma : \Delta_{\mathcal{Y}} \rightrightarrows \mathcal{R}$ if*

$$\forall p \in \Delta_{\mathcal{Y}}, \quad \Gamma(p) = \arg\min_{r \in \mathcal{R}} \langle p, L(r) \rangle \ . \tag{1}$$

*As $\Gamma$ is uniquely defined by $L$, we write $\mathrm{prop}[L]$ to refer to the property elicited by a loss $L$.*

For finite properties (those with $|\mathcal{R}| < \infty$) and discrete losses, we will use lowercase notation $\gamma$ and $\ell$, respectively, with reports $r \in \mathcal{R}$; for surrogate properties and losses we use $\Gamma$ and $L$, with reports $u \in \mathbb{R}^d$. For general properties and losses, we will also use $\Gamma$ and $L$, as above.

## 2.3 Links and Embeddings

To assess whether a surrogate and link function align with the original loss, we turn to the common condition of *calibration*. Roughly, a surrogate and link are calibrated if the best possible expected loss achieved by linking to an incorrect report is strictly suboptimal.

**Definition 4.** *Let original loss $\ell : \mathcal{R} \to \mathbb{R}_+^{\mathcal{Y}}$, proposed surrogate $L : \mathbb{R}^d \to \mathbb{R}_+^{\mathcal{Y}}$, and link function $\psi : \mathbb{R}^d \to \mathcal{R}$ be given. We say $(L, \psi)$ is* calibrated *with respect to $\ell$ if for all $p \in \Delta_{\mathcal{Y}}$,*

$$\inf_{u \in \mathbb{R}^d : \psi(u) \notin \gamma(p)} \langle p, L(u) \rangle > \inf_{u \in \mathbb{R}^d} \langle p, L(u) \rangle \ . \tag{2}$$

It is well-known that calibration implies *consistency*, in the following sense (cf. [2]). Given feature space $\mathcal{X}$, fix a distribution $D \in \Delta(\mathcal{X} \times \mathcal{Y})$. Let $L^*$ be the best possible expected $L$-loss achieved by any hypothesis $H : \mathcal{X} \to \mathbb{R}^d$, and $\ell^*$ the best expected $\ell$-loss for any hypothesis $h : \mathcal{X} \to \mathcal{R}$, respectively. Then $(L, \psi)$ is consistent if a sequence of surrogate hypotheses $H_1, H_2, \ldots$ whose $L$-loss limits to $L^*$, then the $\ell$-loss of $\psi \circ H_1, \psi \circ H_2, \ldots$ limits to $\ell^*$. As Definition 4 does not involve the feature space $\mathcal{X}$, we will drop it for the remainder of the paper.

Several consistent convex surrogates in the literature can be thought of as "embeddings", wherein one maps the discrete reports to a vector space, and finds a convex loss which agrees with the original loss. A key condition is that the original reports should be optimal exactly when the corresponding embedded points are optimal. We formalize this notion as follows.

**Definition 5.** *A loss $L : \mathbb{R}^d \to \mathbb{R}^{\mathcal{Y}}$* embeds *a loss $\ell : \mathcal{R} \to \mathbb{R}^{\mathcal{Y}}$ if there exists some injective embedding $\varphi : \mathcal{R} \to \mathbb{R}^d$ such that (i) for all $r \in \mathcal{R}$ we have $L(\varphi(r)) = \ell(r)$, and (ii) for all $p \in \Delta_{\mathcal{Y}}, r \in \mathcal{R}$ we have*

$$r \in \mathrm{prop}[\ell](p) \iff \varphi(r) \in \mathrm{prop}[L](p) \ . \tag{3}$$

Note that it is not clear if embeddings give rise to calibrated links; indeed, apart from mapping the embedded points back to their original reports via $\psi(\varphi(r)) = r$, how to map the remaining values is far from clear. We address the question of when embeddings lead to calibrated links in Section 4.

To illustrate the idea of embedding, let us examine hinge loss in detail as a surrogate for 0-1 loss for binary classification. Recall that we have $\mathcal{R} = \mathcal{Y} = \{-1, +1\}$, with $L_{\mathrm{hinge}}(u)_y = (1 - uy)_+$ and $\ell_{0\text{-}1}(r)_y := \mathbb{1}\{r \neq y\}$, typically with link function $\psi(u) = \mathrm{sgn}(u)$. We will see that hinge loss embeds (2 times) 0-1 loss, via the embedding $\varphi(r) = r$. For condition (i), it is straightforward

to check that $L_{\text{hinge}}(r)_y = 2\ell_{0\text{-}1}(r)_y$ for all $r, y \in \{-1, 1\}$. For condition (ii), let us compute the property each loss elicits, i.e., the set of optimal reports for each $p$:

$$\text{prop}[\ell_{0\text{-}1}](p) = \begin{cases} 1 & p_1 > 1/2 \\ \{-1, 1\} & p_1 = 1/2 \\ -1 & p_1 < 1/2 \end{cases} \qquad \text{prop}[L_{hinge}](p) = \begin{cases} [1, \infty) & p_1 = 1 \\ 1 & p_1 \in (1/2, 1) \\ [-1, 1] & p_1 = 1/2 \\ -1 & p_1 \in (0, 1/2) \\ (-\infty, -1] & p_1 = 0 \end{cases}.$$

In particular, we see that $-1 \in \text{prop}[\ell_{0\text{-}1}](p) \iff p_1 \in [0, 1/2] \iff -1 \in \text{prop}[L_{\text{hinge}}](p)$, and $1 \in \text{prop}[\ell_{0\text{-}1}](p) \iff p_1 \in [1/2, 1] \iff 1 \in \text{prop}[L_{\text{hinge}}](p)$. With both conditions of Definition 5 satisfied, we conclude that $L_{\text{hinge}}$ embeds $2\ell_{0\text{-}1}$. In this particular case, it is known $(L_{\text{hinge}}, \psi)$ is calibrated for $\psi(u) = \text{sgn}(u)$; in Section 4 we show that, perhaps surprisingly, all embeddings lead to calibration with an appropriate link.

## 3 Embeddings and Polyhedral Losses

In this section, we establish a tight relationship between the technique of embedding and the use of polyhedral (piecewise-linear convex) surrogate losses. We defer to the following section the question of when such surrogates are consistent.

To begin, we observe that our embedding condition in Definition 5 is equivalent to merely matching Bayes risks. This useful fact will drive many of our results.

**Proposition 1.** *A loss $L$ embeds discrete loss $\ell$ if and only if $\underline{L} = \underline{\ell}$.*

*Proof.* Throughout we have $L : \mathbb{R}^d \to \mathbb{R}_+^{\mathcal{Y}}, \ell : \mathcal{R} \to \mathbb{R}_+^{\mathcal{Y}}$, and define $\Gamma = \text{prop}[L]$ and $\gamma = \text{prop}[\ell]$. Suppose $L$ embeds $\ell$ via the embedding $\varphi$. Letting $\mathcal{U} := \varphi(\mathcal{R})$, define $\gamma' : \Delta_{\mathcal{Y}} \rightrightarrows \mathcal{U}$ by $\gamma' : p \mapsto \Gamma(p) \cap \mathcal{U}$. To see that $\gamma'(p) \neq \emptyset$ for all $p \in \Delta_{\mathcal{Y}}$, note that by the definition of $\gamma$ as the property elicited by $\ell$ we have some $r \in \gamma(p)$, and by the embedding condition (3), $\varphi(r) \in \Gamma(p)$. By [9, Lemma 3], we see that $L|_{\mathcal{U}}$ (the loss $L$ with reports restricted to $\mathcal{U}$) elicits $\gamma'$ and $\underline{L} = \underline{L|_{\mathcal{U}}}$. As $L(\varphi(\cdot)) = \ell(\cdot)$ by the embedding, we have

$$\underline{\ell}(p) = \min_{r \in \mathcal{R}} \langle p, \ell(r) \rangle = \min_{r \in \mathcal{R}} \langle p, L(\varphi(r)) \rangle = \min_{u \in \mathcal{U}} \langle p, L(u) \rangle = \underline{L|_{\mathcal{U}}},$$

for all $p \in \Delta_{\mathcal{Y}}$. Combining with the above, we now have $\underline{L} = \underline{\ell}$.

For the reverse implication, assume that $\underline{L} = \underline{\ell}$. In what follows, we implicitly work in the affine hull of $\Delta_{\mathcal{Y}}$, so that interiors are well-defined, and $\underline{\ell}$ may be differentiable on the (relative) interior of $\Delta_{\mathcal{Y}}$. Since $\ell$ is discrete, $-\underline{\ell}$ is polyhedral as the pointwise maximum of a finite set of linear functions. The projection of its epigraph $E_\ell$ onto $\Delta_{\mathcal{Y}}$ forms a power diagram by [3], whose cells are full-dimensional and correspond to the level sets $\gamma_r$ of $\gamma = \text{prop}[\ell]$.

For each $r \in \mathcal{R}$, let $p_r$ be a distribution in the interior of $\gamma_r$, and let $u_r \in \Gamma(p)$. Observe that, by definition of the Bayes risk and $\Gamma$, for all $u \in \mathbb{R}^d$ the hyperplane $v \mapsto \langle v, -L(u_r) \rangle$ supports the epigraph $E_L$ of $-\underline{L}$ at the point $(p, -\langle p, L(u) \rangle)$ if and only if $u \in \Gamma(p)$. Thus, the hyperplane $v \mapsto \langle v, -L(u_r) \rangle$ supports $E_L = E_\ell$ at the point $(p_r, -\langle p_r, L(u_r) \rangle)$, and thus does so at the entire facet $\{(p, -\langle p, L(u_r) \rangle) : p \in \gamma_r\}$; by the above, $u_r \in \Gamma(p)$ for all such distributions as well. We conclude that $u_r \in \Gamma(p) \iff p \in \gamma_r \iff r \in \gamma(p)$, satisfying condition (3) for $\varphi : r \mapsto u_r$. To see that the loss values match, we merely note that the supporting hyperplanes to the facets of $E_L$ and $E_\ell$ are the same, and the loss values are uniquely determined by the supporting hyperplane. (In particular, if $h$ supports the facet corresponding to $\gamma_r$, we have $\ell(r)_y = L(u_r)_y = h(\delta_y)$, where $\delta_y$ is the point distribution on outcome $y$.) $\qquad\square$

From this more succinct embedding condition, we can in turn simplify the condition that a loss embeds *some* discrete loss: it does if and only if its Bayes risk is polyhedral. (We say a concave function is polyhedral if its negation is a polyhedral convex function.) Note that the Bayes risk, a function from distributions over $\mathcal{Y}$ to the reals, may be polyhedral even if the loss itself is not.

**Proposition 2.** *A loss $L$ embeds a discrete loss if and only if $\underline{L}$ is polyhedral.*

*Proof.* If $L$ embeds $\ell$, Proposition 1 gives us $\underline{L} = \underline{\ell}$, and its proof already argued that $\underline{\ell}$ is polyhedral. For the converse, let $\underline{L}$ be polyhedral; we again examine the proof of Proposition 1. The projection of $\underline{L}$ onto $\Delta_{\mathcal{Y}}$ forms a power diagram by [3] with finitely many cells $C_1, \ldots, C_k$, which we can index by $\mathcal{R} := \{1, \ldots, k\}$. Defining the property $\gamma : \Delta_{\mathcal{Y}} \rightrightarrows \mathcal{R}$ by $\gamma_r = C_r$ for $r \in \mathcal{R}$, we see that the same construction gives us points $u_r \in \mathbb{R}^d$ such that $u_r \in \Gamma(p) \iff r \in \gamma(p)$. Defining $\ell : \mathcal{R} \to \mathbb{R}_+^{\mathcal{Y}}$ by $\ell(r) = L(u_r)$, the same proof shows that $L$ embeds $\ell$. $\qquad\square$

Combining Proposition 2 with the observation that polyhedral losses have polyhedral Bayes risks [9, Lemma 5], we obtain the first direction of our equivalence between polyhedral losses and embedding.

**Theorem 1.** *Every polyhedral loss $L$ embeds a discrete loss.*

We now turn to the reverse direction: which discrete losses are embedded by some polyhedral loss? Perhaps surprisingly, we show that *every* discrete loss is embeddable, using a construction via convex conjugate duality which has appeared several times in the literature (e.g. [1, 8, 11]). Note however that the number of dimensions $d$ required could be as large as $|\mathcal{Y}|$.

**Theorem 2.** *Every discrete loss $\ell$ is embedded by a polyhedral loss.*

*Proof.* Let $n = |\mathcal{Y}|$, and let $C : \mathbb{R}^n \to \mathbb{R}$ be given by $(-\underline{\ell})^*$, the convex conjugate of $-\underline{\ell}$. From standard results in convex analysis, $C$ is polyhedral as $-\underline{\ell}$ is, and $C$ is finite on all of $\mathbb{R}^{\mathcal{Y}}$ as the domain of $-\underline{\ell}$ is bounded [27, Corollary 13.3.1]. Note that $-\underline{\ell}$ is a closed convex function, as the infimum of affine functions, and thus $(-\underline{\ell})^{**} = -\underline{\ell}$. Define $L : \mathbb{R}^n \to \mathbb{R}^{\mathcal{Y}}$ by $L(u) = C(u)\mathbb{1} - u$, where $\mathbb{1} \in \mathbb{R}^{\mathcal{Y}}$ is the all-ones vector. We first show that $L$ embeds $\ell$, and then establish that the range of $L$ is in fact $\mathbb{R}_+^{\mathcal{Y}}$, as desired.

We compute Bayes risks and apply Proposition 1 to see that $L$ embeds $\ell$. For any $p \in \Delta_{\mathcal{Y}}$, we have

$$
\begin{aligned}
\underline{L}(p) &= \inf_{u \in \mathbb{R}^n} \langle p, C(u)\mathbb{1} - u \rangle \\
&= \inf_{u \in \mathbb{R}^n} C(u) - \langle p, u \rangle \\
&= - \sup_{u \in \mathbb{R}^n} \langle p, u \rangle - C(u) \\
&= -C^*(p) = -(-\underline{\ell}(p))^{**} = \underline{\ell}(p) \,.
\end{aligned}
$$

It remains to show $L(u)_y \geq 0$ for all $u \in \mathbb{R}^n$, $y \in \mathcal{Y}$. Letting $\delta_y \in \Delta_{\mathcal{Y}}$ be the point distribution on outcome $y \in \mathcal{Y}$, we have for all $u \in \mathbb{R}^n$, $L(u)_y \geq \inf_{u' \in \mathbb{R}^n} L(u')_y = \underline{L}(\delta_y) = \underline{\ell}(\delta_y) \geq 0$, where the final inequality follows from the nonnegativity of $\ell$. $\qquad\square$

## 4   Consistency via Calibrated Links

We have now seen the tight relationship between polyhedral losses and embeddings; in particular, every polyhedral loss embeds some discrete loss. The embedding itself tells us how to link the embedded points back to the discrete reports (map $\varphi(r)$ to $r$), but it is not clear when this link can be extended to the remaining reports, and whether such a link can lead to consistency. In this section, we give a construction to generate calibrated links for *any* polyhedral loss.

The full version [9, Appendix D] contains the full proof; this section provides a sketch along with the main construction and result. The first step is to give a link $\psi$ such that exactly minimizing expected surrogate loss $L$, followed by applying $\psi$, always exactly minimizes expected original loss $\ell$. The existence of such a link is somewhat subtle, because in general some point $u$ that is far from any embedding point can minimize expected loss for two very different distributions $p, p'$, making it unclear whether there exists a choice $\psi(u) \in \mathcal{R}$ that is $\ell$-optimal for both distributions. We show that as we vary $p$ over $\Delta_{\mathcal{Y}}$, there are only finitely many sets of the form $U = \arg\min_{u \in \mathbb{R}^d} \langle p, L(u) \rangle$ [9, Lemma 4]. Associating each $U$ with $R_U \subseteq \mathcal{R}$, the set of reports whose embedding points are in $U$, we enforce that all points in $U$ link to some report in $R_U$. (As a special case, embedding points must link to their corresponding reports.) Proving that these choices are well-defined uses a chain of arguments involving the Bayes risk, ultimately showing that if $u$ lies in multiple such sets $U$, the corresponding report sets $R_U$ all intersect at some $r =: \psi(u)$.

Intuitively, to ensure calibration, we just need to "thicken" this construction, by mapping all approximately-optimal points $u$ to optimal reports $r$. Let $\mathcal{U}$ contain all optimal report sets $U$

of the form above. A key step in the following definition will be to narrow down a "link envelope" $\Psi$ where $\Psi(u)$ denotes the legal or valid choices for $\psi(u)$.

**Definition 6.** *Given a polyhedral $L$ that embeds some $\ell$, an $\epsilon > 0$, and a norm $\|\cdot\|$, the $\epsilon$-thickened link $\psi$ is constructed as follows. First, initialize $\Psi : \mathbb{R}^d \rightrightarrows \mathcal{R}$ by setting $\Psi(u) = \mathcal{R}$ for all $u$. Then for each $U \in \mathcal{U}$, for all points $u$ such that $\inf_{u^* \in U} \|u^* - u\| < \epsilon$, update $\Psi(u) = \Psi(u) \cap R_U$. Finally, define $\psi(u) \in \Psi(u)$, breaking ties arbitrarily. If $\Psi(u)$ became empty, then leave $\psi(u)$ undefined.*

**Theorem 3.** *Let $L$ be polyhedral, and $\ell$ the discrete loss it embeds from Theorem 1. Then for small enough $\epsilon > 0$, the $\epsilon$-thickened link $\psi$ is well-defined and, furthermore, is a calibrated link from $L$ to $\ell$.*

*Sketch. Well-defined:* For the initial construction above, we argued that if some collection such as $U, U', U''$ overlap at a $u$, then their report sets $R_U, R_{U'}, R_{U''}$ also overlap, so there is a valid choice $r = \psi(u)$. Now, we thicken all sets $U \in \mathcal{U}$ by a small enough $\epsilon$; it can be shown that if the *thickened* sets overlap at $u$, then $U, U', U''$ themselves overlap, so again $R_U, R_{U'}, R_{U''}$ overlap and there is a valid chioce $r = \psi(u)$.

*Calibrated:* By construction of the thickened link, if $u$ maps to an incorrect report, i.e. $\psi(u) \notin \gamma(p)$, then $u$ must have at least distance $\epsilon$ to the optimal set $U$. We then show that the minimal gradient of the expected loss along any direction away from $U$ is lower-bounded, giving a constant excess expected loss at $u$. $\qquad\square$

Note that the construction given above in Definition 6 is not necessarily computationally efficient as the number of labels $n$ grows. In practice this potential inefficiency is not typically a concern, as the family of losses typically has some closed form expression in terms of $n$, and thus the construction can proceed at the symbolic level. We illustrate this formulaic approach in § 5.1.

# 5    Application to Specific Surrogates

Our results give a framework to construct consistent surrogates and link functions for any discrete loss, but they also provide a way to verify the consistency or inconsistency of given surrogates. Below, we illustrate the power of this framework with specific examples from the literature, as well as new examples. In some cases we simplify existing proofs, while in others we give new results, such as a new calibrated link for abstain loss, and the inconsistency of the recently proposed Lovász hinge.

## 5.1    Consistency of abstain surrogate and link construction

In classification settings with a large number of labels, several authors consider a variant of classification, with the addition of a "reject" or *abstain* option. For example, Ramaswamy et al. [25] study the loss $\ell_\alpha : [n] \cup \{\bot\} \to \mathbb{R}_+^{\mathcal{Y}}$ defined by $\ell_\alpha(r)_y = 0$ if $r = y$, $\alpha$ if $r = \bot$, and $1$ otherwise. Here, the report $\bot$ corresponds to "abstaining" if no label is sufficiently likely, specifically, if no $y \in \mathcal{Y}$ has $p_y \geq 1 - \alpha$. Ramaswamy et al. [25] provide a polyhedral surrogate for $\ell_\alpha$, which we present here for $\alpha = 1/2$. Letting $d = \lceil \log_2(n) \rceil$ their surrogate is $L_{1/2} : \mathbb{R}^d \to \mathbb{R}_+^{\mathcal{Y}}$ given by

$$L_{1/2}(u)_y = \left( \max_{j \in [d]} B(y)_j u_j + 1 \right)_+ , \tag{4}$$

where $B : [n] \to \{-1, 1\}^d$ is a arbitrary injection; let us assume $n = 2^d$ so that we have a bijection. Consistency is proven for the following link function,

$$\psi(u) = \begin{cases} \bot & \min_{i \in [d]} |u_i| \leq 1/2 \\ B^{-1}(\mathrm{sgn}(-u)) & \text{otherwise} \end{cases} . \tag{5}$$

In light of our framework, we can see that $L_{1/2}$ is an excellent example of an embedding, where $\varphi(y) = B(y)$ and $\varphi(\bot) = 0 \in \mathbb{R}^d$. Moreover, the link function $\psi$ can be recovered from Theorem 3 with norm $\|\cdot\|_\infty$ and $\epsilon = 1/2$; see Figure 1(L). Hence, our framework would have simplified the process of finding such a link, and the corresponding proof of consistency. To illustrate this point further, we give an alternate link $\psi_1$ corresponding to $\|\cdot\|_1$ and $\epsilon = 1$, shown in Figure 1(R):

$$\psi_1(u) = \begin{cases} \bot & \|u\|_1 \leq 1 \\ B^{-1}(\mathrm{sgn}(-u)) & \text{otherwise} \end{cases} . \tag{6}$$

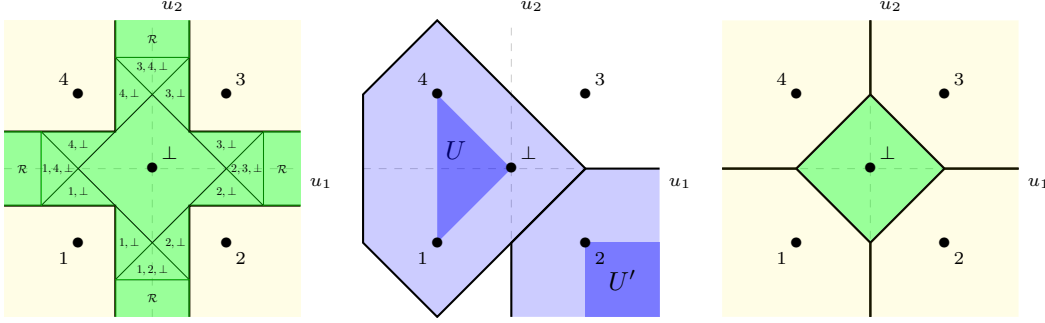

Figure 1: Constructing links for the abstain surrogate $L_{1/2}$ with $d = 2$. The embedding is shown in bold labeled by the corresponding reports. (L) The link envelope $\Psi$ resulting from Theorem 3 using $\|\cdot\|_\infty$ and $\epsilon = 1/2$, and a possible link $\psi$ which matches eq. (5) from [25]. (M) An illustration of the thickened sets from Definition 6 for two sets $U \in \mathcal{U}$, using $\|\cdot\|_1$ and $\epsilon = 1$. (R) The $\Psi$ and $\psi$ from Theorem 3 using $\|\cdot\|_1$ and $\epsilon = 1$.

Theorem 3 immediately gives calibration of $(L_{1/2}, \psi_1)$ with respect to $\ell_{1/2}$. Aside from its simplicity, one possible advantage of $\psi_1$ is that it appears to yield the same constant in generalization bounds as $\psi$, yet assigns $\perp$ to much less of the surrogate space $\mathbb{R}^d$. It would be interesting to compare the two links in practice.

## 5.2 Inconsistency of Lovász hinge

Many structured prediction settings can be thought of as making multiple predictions at once, with a loss function that jointly measures error based on the relationship between these predictions [14, 16, 23]. In the case of $k$ binary predictions, these settings are typically formalized by taking the predictions and outcomes to be $\pm 1$ vectors, so $\mathcal{R} = \mathcal{Y} = \{-1, 1\}^k$. One then defines a joint loss function, which is often merely a function of the set of mispredictions, meaning we may write $\ell^g(r)_y = g(\{i \in [k] : r_i \neq y_i\})$ for some set function $g : 2^{[k]} \to \mathbb{R}$. For example, Hamming loss is given by $g(S) = |S|$. In an effort to provide a general convex surrogate for these settings when $g$ is a submodular function, Yu and Blaschko [32] introduce the *Lovász hinge*, which leverages the well-known convex Lovász extension of submodular functions. While the authors provide theoretical justification and experiments, consistency of the Lovász hinge is left open, which we resolve.

Rather than formally define the Lovász hinge, we defer the complete analysis to the full version of the paper [9], and focus here on the $k = 2$ case. For brevity, we write $g_\emptyset := g(\emptyset)$, $g_{1,2} := g(\{1, 2\})$, etc. Assuming $g$ is normalized and increasing (meaning $g_{1,2} \geq \{g_1, g_2\} \geq g_\emptyset = 0$), the Lovász hinge $L : \mathbb{R}^k \to \mathbb{R}_+^{\mathcal{Y}}$ is given by

$$L^g(u)_y = \max\Big\{(1 - u_1 y_1)_+ g_1 + (1 - u_2 y_2)_+ (g_{1,2} - g_1),$$
$$(1 - u_2 y_2)_+ g_2 + (1 - u_1 y_1)_+ (g_{1,2} - g_2)\Big\}, \quad (7)$$

where $(x)_+ = \max\{x, 0\}$. We will explore the range of values of $g$ for which $L^g$ is consistent, where the link function $\psi : \mathbb{R}^2 \to \{-1, 1\}^2$ is fixed as $\psi(u)_i = \mathrm{sgn}(u_i)$, with ties broken arbitrarily.

Let us consider the coefficients $g_\emptyset = 0$, $g_1 = g_2 = g_{1,2} = 1$, for which $\ell^g$ is merely 0-1 loss on $\mathcal{Y}$. For consistency, for any distribution $p \in \Delta_{\mathcal{Y}}$, we must have that whenever $u \in \arg\min_{u' \in \mathbb{R}^2}\langle p, L^g(u')\rangle$, the outcome $\psi(u)$ must be the most likely, i.e., in $\arg\max_{y \in \mathcal{Y}} p(y)$. Simplifying eq. (7), however, we have

$$L^g(u)_y = \max\{(1 - u_1 y_1)_+, (1 - u_2 y_2)_+\} = \max\{1 - u_1 y_1, 1 - u_2 y_2, 0\}, \quad (8)$$

which is exactly the abstain surrogate (4) for $d = 2$. We immediately conclude that $L^g$ cannot be consistent with $\ell^g$, as the origin will be the unique optimal report for $L^g$ under distributions with $p_y < 0.5$ for all $y$, and one can simply take a distribution which disagrees with the way ties are broken in $\psi$. For example, if we take $\mathrm{sgn}(0) = 1$, then under $p((1,1)) = p((1,-1)) = p((-1,1)) = 0.2$ and $p((-1,-1)) = 0.4$, we have $\{0\} = \arg\min_{u \in \mathbb{R}^2}\langle p, L^g(u)\rangle$, yet we also have $\psi(0) = (1,1) \notin \{(-1,-1)\} = \arg\min_{r \in \mathcal{R}}\langle p, \ell^g(r)\rangle$.

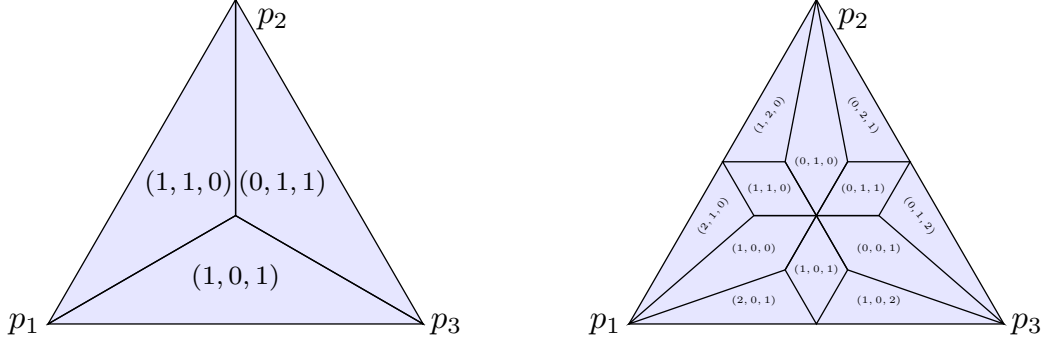

Figure 2: Minimizers of $\langle p, \ell^{\text{top-2}} \rangle$ and $\langle p, \ell^2 \rangle$, respectively, varying $p$ over $\Delta_3$.

In fact, this example is typical: using our embedding framework, and characterizing when $0 \in \mathbb{R}^2$ is an embedded point, one can show that $L^g$ is consistent if and only if $g_{1,2} = g_1 + g_2$. Moreover, in this linear case, which corresponds to $g$ being *modular*, the Lovász hinge reduces to weighted Hamming loss, which is trivially consistent from the consistency of hinge loss for 0-1 loss. In the full version of the paper [9], we generalize this observation for all $k$: $L^g$ is consistent if and only if $g$ is modular. In other words, even for $k > 2$, the only consistent Lovász hinge is weighted Hamming loss. These results cast doubt on the effectiveness of the Lovász hinge in practice.

## 5.3   Inconsistency of top-$k$ losses

In certain classification problems when ground truth may be ambiguous, such as object identification, it is common to predict a set of possible labels. As one instance, the top-$k$ classification problem is to predict the set of $k$ most likely labels; formally, we have $\mathcal{R} := \{r \in \{0,1\}^n : \|r\|_0 = k\}$, $1 < k < n$, $\mathcal{Y} = [n]$, and discrete loss $\ell^{\text{top-}k}(r)_y = 1 - r_y$. Surrogates for this problem commonly take reports $u \in \mathbb{R}^n$, with the link $\psi(u) = \{u_{[1]}, \ldots, u_{[k]}\}$, where $u_{[i]}$ is the $i^{th}$ largest entry of $u$.

Lapin et al. [19, 20, 21] provide the following convex surrogate loss for this problem, which Yang and Koyejo [31] show to be inconsistent:[1]

$$L^k(u)_y := \left( 1 - u_y + \tfrac{1}{k} \sum_{i=1}^{k} (u - e_y)_{[i]} \right)_+ , \tag{9}$$

where $e_y$ is 1 in component $y$ and 0 elsewhere. With our framework, we can say more. Specifically, while $(L^k, \psi)$ is not consistent for $\ell^{\text{top-}k}$, since $L^k$ is polyhedral, we know from Theorem 1 that it embeds *some* discrete loss $\ell^k$, and from Theorem 3 there is a link $\psi'$ such that $(L^k, \psi')$ is calibrated (and consistent) for $\ell^k$. We therefore turn to deriving this discrete loss $\ell^k$.

For concreteness, consider the case with $k = 2$ over $n = 3$ outcomes. We can re-write $L^2(u)_y = \left( 1 - u_y + \tfrac{1}{2}(u_{[1]} + u_{[2]} - \min(1, u_y)) \right)_+$. By inspection, we can derive the properties elicited by $\ell^{\text{top-2}}$ and $L^2$, respectively, which reveals that the set $\mathcal{R}'$ consisting of all permutations of $(1,0,0)$, $(1,1,0)$, and $(2,1,0)$, are always represented among the minimizers of $L^2$. Thus, $L^2$ embeds the loss $\ell^2(r)_y = 0$ if $r_y = 2$ or $\ell^2(r)_y = 1 - r_y + \tfrac{1}{2}\langle r, \mathbb{1} - e_y \rangle$ otherwise. Observe that $\ell^2$ is just $\ell^{\text{top-2}}$ with an extra term punishing weight on elements other than $y$, and a reward for a weight of 2 on $y$.

Moreover, we can visually inspect the corresponding properties (Fig. 2) to immediately see why $L^2$ is inconsistent: for distributions where the two least likely labels are roughly equally (un)likely, the minimizer will put all weight on the most likely label, and thus fail to distinguish the other two. More generally, $L^2$ cannot be consistent because the property it embeds does not "refine" (subdivide) the top-$k$ property, so not just $\psi$, but *no* link function, could make $L^2$ consistent.

# 6  Conclusion and Future Directions

This paper formalizes an intuitive way to design convex surrogate losses for classification-like problems—by embedding the reports into $\mathbb{R}^d$. We establish a close relationship between embeddings and polyhedral surrogates, showing both that every polyhedral loss embeds a discrete loss (Theorem 1) and that every discrete loss is embedded by some polyhedral loss (Theorem 2). We then construct a calibrated link function from any polyhedral loss to the discrete loss it embeds, giving consistency for all such losses (Theorem 3). We conclude with examples of how the embedding framework presented can be applied to understand existing surrogates in the literature, including those for the abstain loss, top-$k$ loss, and Lovász hinge. In particular, our link construction recovers the link function proposed by Ramaswamy et al. [25] for abstain loss, as well as another simpler link based on the $L_1$ norm.

One open question of particular interest involves the dimension of the surrogate prediction space; given a discrete loss, can we construct a surrogate that embeds it *of minimal dimension*? If we naïvely embed the reports into an $n$-dimensional space, the dimensionality of the problem scales linearly in the number of possible labels $n$. As the dimension of the optimization problem is a function of this *embedding dimension $d$*, a promising direction is to leverage tools from elicitation complexity [13, 18] and convex calibration dimension [24] to understand when we can take $d \ll n$.

## Acknowledgements

We thank Arpit Agarwal and Peter Bartlett for many early discussions, which led to several important insights. We thank Eric Balkanski for help with a lemma about submodular functions. This material is based upon work supported by the National Science Foundation under Grant No. 1657598.

## Footnotes

[1]Yang and Koyejo also introduce a consistent surrogate, but it is non-convex.

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
