[Reviews · NeurIPS 2019]

Reviewer 1



The paper is very well written, making the sometimes dry topic of losses easy to follow and engaging. I want to specially mention that the authors did a great job of providing intuitive interpretations of most definitions, which will surely be appreciated by many readers. However I think that there's room for improvement in further clarifying terms like "reports" and "properties" which will most likely be new for a significant part of the audience. In terms of the technical contribution, it provides a fresh look at the construction of surrogate losses from the point of view of embeddings, proving the interesting result that every discrete loss can be embedded into a polyhedral loss as well as the converse. On top of the conceptual appeal, the power of this idea is exemplified by reworking or deriving new results on consistency and calibration of surrogate losses. Proofs are solid and generally the paper is a valuable addition to the literature on losses and surrogates.

Reviewer 2



This work considers the relationship between convex surrogate loss and learning problem such as classification and ranking. The authors embed each of the finitely many predictions (e.g. classes) as a point in Rd, assign the original loss values to these points, and convexifies the loss in between to obtain a surrogate. The authors prove that this approach is equivalent, in a strong sense, to working with polyhedral (piecewise linear convex) losses, and give a construction of a link function through which L is a consistent surrogate for the loss it embeds. Some examples are presented to verify the theoretical analysis. This is an interesting direction in learning theory, while I have some concerns as follows: 1) What's the motivation of polyhedral losses? The authors should present some real applications and shows its importance, especially for some new learning problems and settings. 2) It would be better that the authors focus on some specific learning setting such as classification, or AUC or classification with rejection. The presented submission focuses on various setting and there is a lack of deep understanding on some specific problems. 3) It would be better to present some intuitive explanation for Definitions 1-3 for better understanding their meanings. 4) There are some known results in Section 5 and the authors should present some new real applications of surrogate loss. 5) It would be better to present some regret bounds on the surrogate loss and polyhedral losses.

Reviewer 3



********After author response*********************** I thank the authors for answering my questions. I keep my evaluation and vote for accepting the paper. ********************************************************** Originality: This work provided a novel approach for designing convex loss surrogates for general discrete losses and analyzing their consistency. Using the new analysis framework, the authors gave a negative answer to the open question of the consistency of the Lovasz hinge. The authors also gave original understandings of the top-k loss, showing that the convex loss by [LHS15] is consistent with a discrete loss that is slightly different from the discrete top-k loss. Quality: I haven't gone through the appendix, but the proofs and arguments in the main paper are sound and clear. Clarity: The paper is very clearly written and easy to understand. Significance: Convex loss surrogates are broadly used in machine learning, so understanding how to design such surrogates systematically is meaningful. The analysis of top-k losses and Lovasz hinge helps practitioners understand these losses better and gives them insights for choosing the right loss. [LHS15] Lapin, Maksim and Hein, Matthias and Schiele, Bernt. Top-k multiclass SVM

[Author Response · NeurIPS 2019]

We thank all reviewers for their constructive feedback. Two of the reviewers suggested additional intuition and
explanation for definitions and terminology (report, property, elicits, etc) which we will address. We would also like to
mention that we found and fixed an error in the top-k analysis; see response to R3. Individual responses follow.

**R1:** Thank you for your comments. As we state above, we will clarify terms like *reports* and *properties*.

**R2:**

Thank you for your comments and questions. We respond to your individual points below, but to address an overall
theme, we would like to emphasize that this is a theory paper. Our goal is to provide a general theoretical framework
which allows practitioners to design new surrogates for new settings without having to do so entirely from scratch. For
example, through our framework there is guaranteed to be a link function which gives consistency, and in most cases
the proof is constructive enough to derive it directly, as we illustrate with the abstain surrogate. We do not, however,
seek to create a new learning problem; when practitioners have a reason to study a new problem, they can apply our
framework to understand their problem better.

1. We feel that the best motivation for polyhedral losses is to enumerate the many examples which appear already in the
literature (hinge, top-k, abstain, Lovász hinge, etc), rather than come up with new settings which may or may not be
of practical interest. Another motivation is the close connection with loss embedding, which is a natural approach to
designing convex surrogates (of any kind).

2. We deliberately chose not to focus on any specific setting, to emphasize the generality of our framework. This choice
does make the paper more abstract, so we adopted several running examples (hinge, abstain) to illustrate the results;
we will look for more places to add such illustration. Finally, in some sense, our results do deepen understanding of
specific settings. For example, we give new intuition for a proposed surrogate for the top-k classification problem, and
the Lovász hinge: why it is not consistent, and more interestingly, for what problem it *is* consistent.

3. As mentioned above, we will provide more intuition for these definitions.

4. While we do not invent new learning settings, as justified above, our work does indeed provide new results for
specific settings, such as for top-k and the Lovász hinge. For the latter, it was previously an open question if the Lovász
hinge was a consistent surrogate for any of the broad array of settings it encompasses aside from Hamming loss – we
show that in fact it is not consistent for any of these settings.

5. We interpreted your comment to mean Bayesian regret (please correct us in the final review if we are mistaken). We
expect that one can prove a general form for such regret bounds, depending on certain parameters of the polyhedral
loss such as the maximum gradient and the minimum distance (in some sense) between embedded points. Given how
complex the analysis is to establish consistency, we have left the challenging question of regret bounds for future work.

**R3**:

Thank you for your comments and questions. First, the top-k correction: The form of the discrete loss in eq. (9) should
be slightly different, though the intuition is essentially the same: there is a term for the original top-k discrete loss, plus
a cardinality penalty, plus an additional term which allows one to express higher confidence in some labels than others
(but still from a discrete set). We have corrected the proof and exposition.

Regarding your questions:

1. Excellent question; we will add a discussion in the paper. The polyhedral loss given in Theorem 2 would likely
not be "computed" per se, as the discrete loss typically depends on the number of labels $n$, and one would want a
mathematical expression for the loss in terms of $n$. This expression, which is essentially the Fenchel conjugate of
a polyhedral function, follows from standard results in convex analysis [Rockafellar, 1997, Thm 19.1, Thm 19.2].
Similarly, the link $\psi$ in Theorem 3 would be derived mathematically, which may be challenging in some cases but
typically straightforward, such as the new link we give for abstain loss.

2. The surrogate constructed in Theorem 2 is one consistent surrogate, but takes $2^k$ dimensions. For which problems
this construction is as good as one could hope (i.e., yields the lowest dimensional consistent surrogate), and for which
the dimension could be significantly reduced, is a challenging open question, and the subject of our ongoing work.

# References

R.T. Rockafellar. *Convex analysis*, volume 28 of *Princeton Mathematics Series*. Princeton University Press, 1997.


[Meta-Review · NeurIPS 2019]

Solid theoretical paper on designing convex surrogate losses. Inserting all the comments made to answers R2 and R3 is a must for the paper to be very good.